# Cardiogenic programming of human pluripotent stem cells by dose-controlled activation of EOMES

Martin J. Pfeiffer[1,2,5], Roberto Quaranta[1,2], Ilaria Piccini[1,3], Jakob Fell[1,2], Jyoti Rao[1,2,6], Albrecht Röpke[4], Guiscard Seebohm[3] & Boris Greber [1,2,7]

Master cell fate determinants are thought to induce specific cell lineages in gastrulation by orchestrating entire gene programs. The T-box transcription factor EOMES (eomesodermin) is crucially required for the development of the heart—yet it is equally important for endoderm specification suggesting that it may act in a context-dependent manner. Here, we define an unrecognized interplay between EOMES and the WNT signaling pathway in controlling cardiac induction by using loss and gain-of-function approaches in human embryonic stem cells. Dose-dependent *EOMES* induction alone can fully replace a cocktail of signaling molecules otherwise essential for the specification of cardiogenic mesoderm. Highly efficient cardiomyocyte programming by EOMES mechanistically involves autocrine activation of canonical WNT signaling via the WNT3 ligand, which necessitates a shutdown of this axis at a subsequent stage. Our findings provide insights into human germ layer induction and bear biotechnological potential for the robust production of cardiomyocytes from engineered stem cells.

[1] Human Stem Cell Pluripotency Laboratory, Max Planck Institute for Molecular Biomedicine, 48149 Münster Germany. [2] Chemical Genomics Centre of the Max Planck Society, 44227 Dortmund Germany. [3] Department of Cardiovascular Medicine, Institute of Genetics of Heart Diseases, University of Münster Medical School, 48149 Münster Germany. [4] Institute of Human Genetics, University Hospital Münster, 48149 Münster Germany. [5] Present address: Centre of Reproductive Medicine and Andrology, University Hospital Münster, Münster 48149, Germany. [6] Present address: Department of Genetics, Harvard Medical School and Brigham and Women's Hospital, Boston, MA 02115, USA. [7] Present address: RheinCell Therapeutics GmbH, Boston 40764, USA. Correspondence and requests for materials should be addressed to B.G. (email: b.greber@rheincell.de)

Essentially all heart cells are descendants of *Eomes*-expressing cells in mouse development[1-4]. Similarly, most cells forming the heart are derived from mesoderm precursors expressing the bHLH transcription factor *Mesp1*[5,6]. MESP1 has been proposed to play a master regulatory role in cardiovascular specification[7]. This view is based on procardiac effects observed in elegant MESP1 gain-of-function studies using mouse embryonic stem (ES) cells and vertebrate embryos[7-10]. *Mesp1* is a target gene of EOMES and hence it is thought that EOMES exerts its cardiogenic function through this mechanism[1,11]. However, neither EOMES nor MESP1-expressing cells in the embryo exclusively give rise to the cardiac lineage, since both genes also play prominent roles in other contexts[2,10,12,13]. Accordingly, overexpression studies in mouse ES cells have thus far yielded rather moderate cardiogenic effects over background[7-11]. Therefore, the issue of whether there is a bona fide master regulatory factor specifically promoting the induction of cardiac cells at high efficiency, and under which conditions it would do so, appears to be unresolved.

Human ES cells (hESCs) present an excellent model system to investigate such questions. This is because controlled differentiation procedures, including directed cardiac induction protocols, are in part highly developed by now and these are based on developmental principles[14,15]. In addition, genetic manipulation tools have emerged that now permit systematic loss and gain-of-function studies, in combination with modifying the extrinsic signaling environment at high temporal resolution. Here, we demonstrate that within an intermediate corridor of transcriptional activation, *EOMES* may specifically activate a cardiogenic program in hESCs. This alternative approach of promoting cardiac induction does not require exogenous signaling cascade activation, yet it necessitates an inhibition of the WNT pathway at the cardiac mesoderm stage. Mechanistic investigation establishes that this accessory requirement is based on a functional link between *EOMES* and the *WNT3* locus.

## Results

**EOMES knockout (KO) hESCs do not form cardiomyocytes (CMs).** Following up on our previous investigation of cardiac induction mechanisms in the hESC system[16], we subjected EOMES KO hESCs to a stringent cardiac differentiation protocol[17]. At the cardiac mesoderm stage of this procedure[18], day 2, EOMES was confirmed to be highly expressed in wild-type (WT) cells and absent in KO ones (Fig. 1a). At day 8, WT cells had formed beating monolayers expressing the early cardiomyocyte

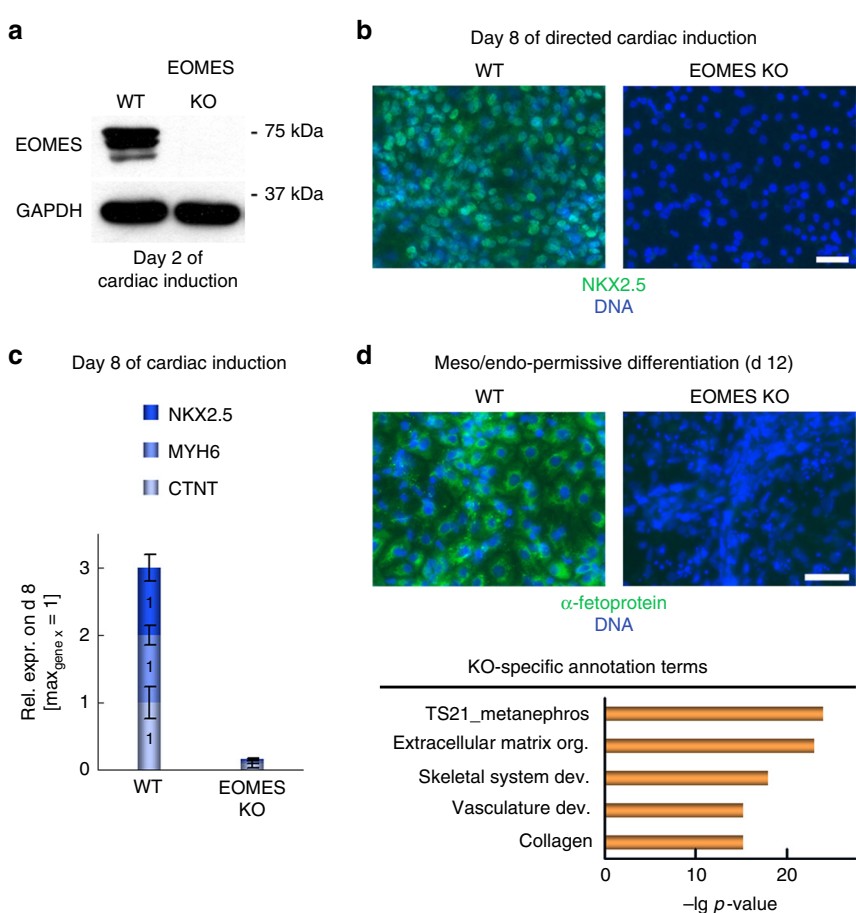

**Fig. 1** EOMES knockout hESCs fail to differentiate into cardiomyocytes. **a** Immunoblot confirming EOMES expression and its absence in WT and KO cells, respectively, at the cardiac mesoderm stage of directed differentiation. **b** EOMES KO cells fail to express the early cardiomyocyte marker NKX2.5 following exposure to a directed differentiation protocol. Scale bar: 50 μm. **c** EOMES KO cells show a general failure in markedly upregulating essential pan-cardiac genes (qPCR data, n = 3; error bars: s.e.m.). For each gene, the expression level in the highest expressing sample ("max" = WT in all three cases) is set to 1. **d** Differentiation of EOMES KO hESCs under signaling factor-assisted, non-cardiac mesoderm/endoderm-permissive culture conditions. Top: EOMES KO cells fail to express α-fetoprotein following Activin A-assisted endodermal induction. Scale bar: 50 μm. Bottom: Non-cardiac mesodermal differentiation competence of EOMES KO hESCs. KO cells were differentiated using varied meso and endoderm-permissive induction conditions. The data denote transformed p-values of annotation terms enriched in gene sets upregulated in EOMES KO samples compared to undifferentiated hESCs or differentiated WT controls (cutoffs: 10 or 3-fold, respectively). Underlying data: Supplementary Data 1

marker NKX2.5 and other pan-cardiac genes, whereas EOMES KO cells did virtually not express any of these (Fig. 1b, c). Likewise, using Activin A-assisted differentiation conditions, WT hESCs readily formed endodermal cells, whereas EOMES KO cells entirely failed to do so, as expected from literature (Fig. 1d, top).

Next, we asked whether *EOMES* deficiency would disable somatic differentiation in general or, at least, globally prevent mesodermal commitment. To this end, WT and EOMES KO hESCs were subjected to spontaneous differentiation conditions or to signaling factor-assisted, non-cardiac mesoderm-permissive ones, as based on a previously established mesendodermal patterning model[16]. EOMES KO cells readily differentiated along the neural lineage (Supplementary Fig. 1a). Moreover, an unbiased expression analysis of meso-permissive differentiation cultures suggested that *EOMES* disruption preserves differentiation competence into renal, mesenchymal, and endothelial lineages (Fig. 1d, bottom, and Supplementary Data 1). Immunofluorescent stainings confirmed the ability of EOMES KO cells to differentiate into these exemplary cell types (Supplementary Fig. 1b). Thus, with regard to mesodermal commitment, EOMES is crucially required for CM formation but not for mesodermal differentiation in general.

**EOMES drives cardiac programming of hESCs at high efficiency.** Given the severe failure of EOMES KO hESCs to form CMs under directed differentiation conditions, we next asked whether enforced *EOMES* induction could in turn drive the process on its own. Using an inducible overexpression cell line on $EOMES^{KO}$ background in which an EOMES transgene may be activated using doxycycline (DOX) administration ($EOMES^{KO/E.TET-ON}$, Supplementary Fig. 2a)[16], we modified the standard cardiac induction procedure such that all cardiac mesoderm-inducing signaling factors were replaced by DOX (Fig. 2a). Optimization of the timing of DOX supplementation and that of a WNT inhibitor added in a subsequent stage suggested that 3 days of DOX treatment combined with 2 days of WNT inhibition (from 48–96 h) was most optimal (Supplementary Fig. 2b). Strikingly, this protocol—devoid of any signaling pathway-activating molecules—generated near-homogeneous monolayers of beating CMs, similar to the standard procedure (Fig. 2b and Supplementary Movie 1).

The hESC-CMs programmed by EOMES were termed pCMs and characterized as follows. pCMs expressed structural markers and ion channel genes involved in forming cardiac action potentials, at similar levels compared to standard hESC-CMs (Supplementary Fig. 2c). Furthermore, pCMs displayed robust

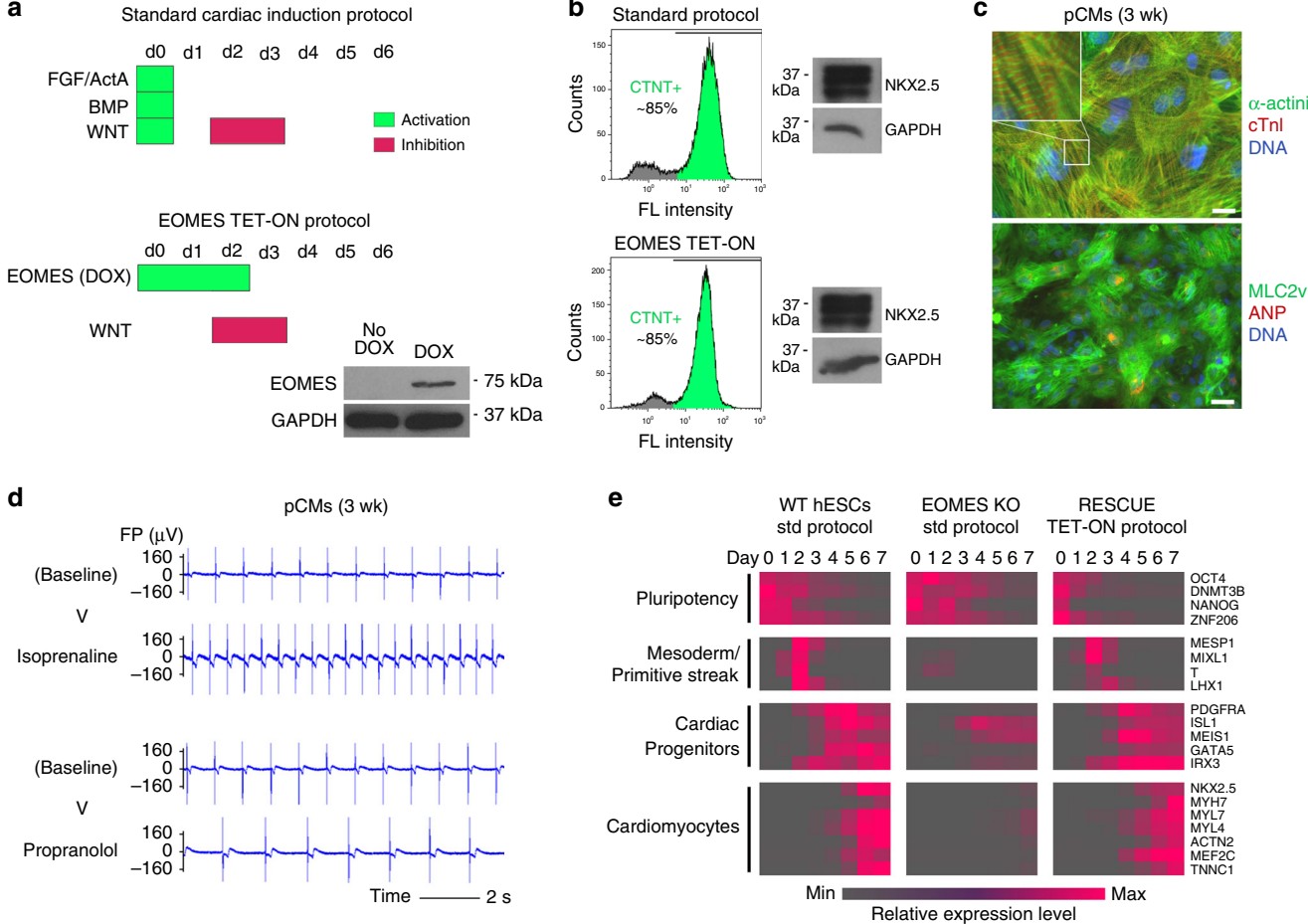

**Fig. 2** EOMES programs hESCs into functional CMs at high efficiency. **a** Illustration of growth factor-mediated and optimized *EOMES* induction-based cardiac differentiation protocols. Bottom right: Immunoblot validating doxycycline-dependent EOMES expression in a transgenic $EOMES^{KO/E.TET-ON}$ hESC line. **b** Typical yields of hESC-CMs (left, flow cytometry) and NKX2.5 expression (right, immunoblot) obtained with the two protocols (day 10). **c** Immunostainings 21 days after the initiation of *EOMES* induction. Weak perinuclear ANP staining is typical in overall MLC2v-positive hESC-CMs. Scale bars: 25 (top) and 50 μm (bottom). **d** Acceleration and slowdown of spontaneous beat rates in pCMs following exposure to 10 μM isoprenaline and 10 μM propranolol, respectively, on multielectrode arrays. **e** Microarray-based time course analysis comparing the indicated protocols and cell lines. RESCUE cells carry an inducible *EOMES* transgene on $EOMES^{KO}$ HuES6 background. Underlying data are from Supplementary Data 2

staining for cardiac markers at the protein level and showed a striated sarcomeric pattern in many cells already by 3 weeks (wk) (Fig. 2b, c). hESC-CMs tend to acquire an overall ventricular subtype identity by default, whereas an atrial fate may be induced by retinoic acid addition during early stages of differentiation[19]. We adopted and employed this paradigm to assess the subtype identity of pCMs. pCMs robustly expressed ventricular-specific myosin light chain 2 in almost all cells and were essentially deficient in atrial-specific ANP expression (Fig. 2c). These and other heart chamber-specific markers suggest that pCMs take on an overall ventricular identity, similar to standard hESC-CMs but different from retinoic acid-treated ones (Supplementary Fig. 2c).

To substantiate this conclusion, and to demonstrate the overall physiological functionality of pCMs, these cells were treated with E-4031, an inhibitor of the hERG potassium channel that plays a key role in CM repolarisation[20]. E-4031 prolonged electrical field potential durations on multielectrode arrays indicating that hERG is operative in pCMs (Supplementary Fig. 2d). Moreover, pCMs did not display significant field potential prolongations upon addition of 4-aminopyridine, an inhibitor of the atrial-specific KCNA5 channel—much in contrast to standard atrial-like CMs generated via transient retinoic acid treatment[21] (Supplementary Fig. 2e). Furthermore, pCMs showed physiological responses to chronotropic drugs, namely, beat rate acceleration in response to isoprenaline, an adrenergic agonist, and slowdown after addition of propranolol which is a beta blocker (Fig. 2d). These data indicate that pCMs compare well with conventional ventricular-like hESC-CMs regarding their overall characteristics and maturation features acquired upon prolonged in vitro culture[17].

**Differentiation fate is *EOMES* dose dependent**. We next sought to better understand cardiac programming by EOMES. A time course gene expression analysis indicated that EOMES KO cells not only failed to eventually upregulate a cardiac program but that

they were already lacking entire sets of mesodermal and primitive streak genes at early differentiation stages. However, *EOMES* induction via the TET-ON protocol fully restored the normal sequence of events (Fig. 2e and Supplementary Data 2). Hence, the similarity between conventional hESC-CMs and pCMs at later stages is matched by highly similar induction kinetics of mesodermal and cardiac gene expression in the short term.

To check whether endogenous *EOMES* influences the efficacy of the transgenic induction system in a negative way, and to further elucidate the requirements for pCM formation, we additionally generated an EOMES TET-ON line on WT hESC background (WT[E.TET-ON] hESCs, Supplementary Fig. 3a, b). There was no apparent difference in cardiac differentiation efficiencies, and in general the system conveniently allowed pCM induction from routine hESC maintenance plates—simply by adding DOX to subconfluent 6-well cultures (Supplementary Movie 2). pCM induction efficiency, however, was highly dependent on the DOX concentration used (Fig. 3a). Moreover, the TET-ON protocol strictly required the inactivation of WNT signaling in a subsequent differentiation stage (Fig. 3b). Outside the pCM-compatible DOX range, the TET-ON protcol interestingly gave rise to alternative mesendodermal differentiation fates. Hence, following low-DOX induction (0.125 μg/ml), these included hepatic as well as putative hematopoetic cells characterized by hemoglobin gene expression. And following high-DOX induction (1 μg/ml), WT[E.TET-ON] hESCs heterogeneously differentiated into smooth muscle and mesenchymal fates (Supplementary Fig. 3c, d). A similar dependence of cardiac induction on the DOX concentration was also noticed in a third cell line generated on an independent induced pluripotent stem cell background[21] (WT[E.TET-ON] hiPSCs, Supplementary Fig. 3e, f): Only at a DOX concentration range of 0.1–0.13 μg/ml did this cell line reproducibly form pCMs at high efficiency (Fig. 3c and Supplementary Movie 3).

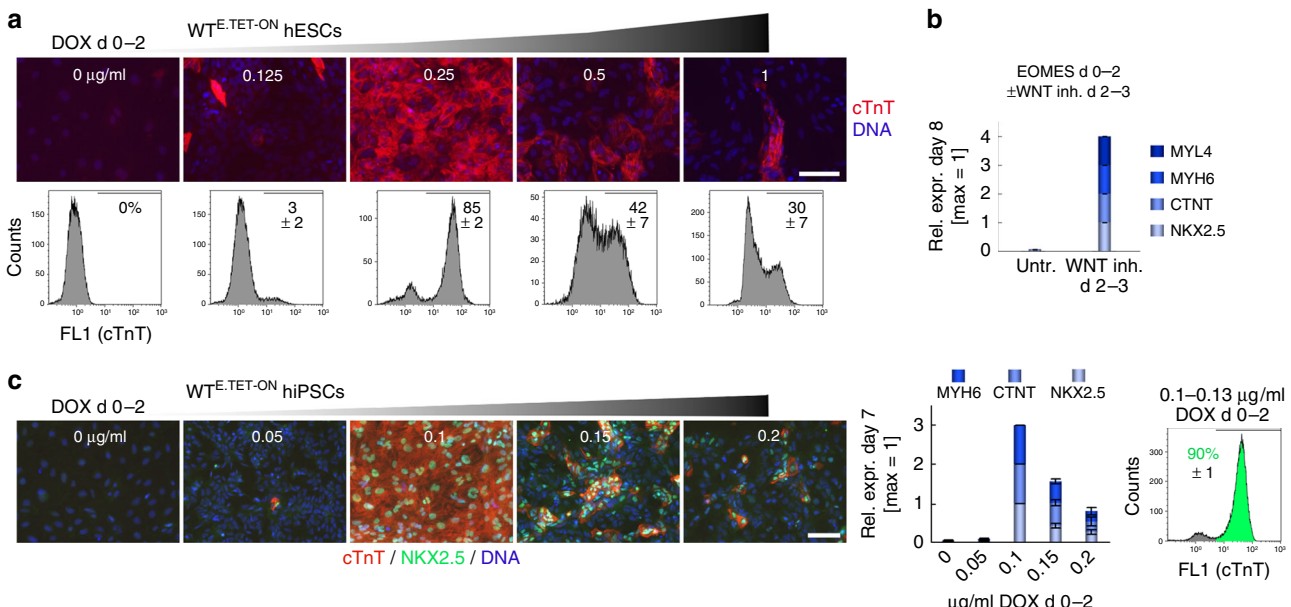

**Fig. 3** Contextual requirements of EOMES-mediated CM programming. **a** DOX dose dependency of the EOMES TET-ON protocol using the WT[E.TET-ON] hESC line. Top panel: Immunostains at 1.5 wk. Scale bar: 100 μm. Bottom: Flow cytometry analysis. Numbers indicate average percentages of cTnT-positive CMs from 3–6 experiments per condition. **b** CM programming by EOMES necessitates suppression of autocrine WNT signaling from the third day of transgene induction (qPCR data, n = 2). **c** DOX dose-dependent CM programming using an independent WT[E.TET-ON] hiPSC line. The data shows immunostaining (scale bar: 100 μm), RT-qPCR (n = 4), and FACS analysis (n = 3) performed at 1 wk of differentiation. The asymmetrical shape of the DOX titration data with this line is in part due to the incomplete repression of SOX2 at 0.05 μg/ml, which caused overgrowth of the cultures with neural precursors (also see Supplementary Fig. 3f). Error bars: s.e.m.

**Cardiogenic activity of *EOMES* is crucially linked to *WNT3*.** This behavior was reminiscent of our standard growth factor-based protocol in which the BMP and/or WNT signaling pathways can easily be under or overstimulated, giving rise to non-cardiac fates in such cases[16]. We hence hypothesized that the profound impact of EOMES on global gene expression programs may in part be based on a link to WNT or BMP signaling. Short-term WNT and BMP stimulation experiments defined pathway-specific target genes in hESCs (Supplementary Data 3). Some of the most stringent ones were then monitored in the TET-ON differentiation time course. This analysis indicated that during the first 2 days of *EOMES* induction, target genes of canonical WNT signaling became strongly upregulated, whereas BMP-linked genes only showed a response thereafter (Fig. 4a). Notably, some of the early-induced WNT targets, like *MSX1* and *CDX1/2*, are known to be counterproductive for cardiac differentiation[16]. Their upregulation explains the necessity for subsequent WNT inhibition and argues for the involvement of a global—not necessarily cardiac-specific—WNT response. As expected, WNT inhibition during the first 2 days of DOX treatment prevented the upregulation of WNT targets including *T* (Brachyury), indicating that EOMES alone was not sufficient for their induction (Supplementary Fig. 4a). Similar treatments also suppressed cardiac differentiation in the longer term, which demonstrates the functional importance of an initial activation of WNT signaling downstream of EOMES (Fig. 4b, top). In comparison, early BMP inhibition only had a mild effect (Fig. 4b, bottom).

To identify a possible mechanism for these observations, the TET-ON differentiation time course data set was mined for the early induction of genes encoding canonical WNT signaling ligands (Supplementary Data 2). Two such genes, *WNT3* and *WNT3A*, were markedly upregulated by *EOMES* overexpression within 1–2 days (Fig. 4c). Both of these were also bound by EOMES in differentiating hESCs, according to a previously published ChIP-seq data set[13]. Using ChIP-qPCR amplicons overlapping with the corresponding promoter regions, we were able to confirm *WNT3*—but not *WNT3A*—as a direct EOMES target gene in our protocol (Fig. 4d). Nonetheless, we sought to investigate the functional implication of both ligand-encoding genes in an unbiased manner. Hence, we prepared homozygous KO hESC lines for both *WNT3* and *WNT3A* using quadruple CRISPR/Cas9n DNA nicking[22]. To account for a potential compensation between the two targeted genes, we also generated a *WNT3/WNT3A* double knockout line (DKO, see Methods and Supplementary Table 1). These lines were prepared on WT[E.TET-ON] background to later be able to challenge WNT ligand function using the TET-ON protocol. The induced splice mutations caused the predicted exon 2 deletions and associated reading frame shifts at the RNA level, as evidenced by exon-spanning RT-PCR and cDNA sequencing (Fig. 4e and Supplementary Fig. 4b).

DOX supplementation for 2 days induced *SP5*, a stringent WNT-specific target gene in hESCs, only in WT[E.TET-ON] cells as well as in the WNT3A KO line, but not in WNT3 KO or WNT3/WNT3A DKO cells (Supplementary Fig. 4c). These data indicate that WNT3, not WNT3A, mediates the EOMES-induced WNT response early in the protocol. Furthermore, they predicted that CM formation would be compromised in the *WNT3* mutant lines. Indeed, cardiac differentiation efficiencies were unaffected in WNT3A KO cells but severely diminished in WNT3 KO and WNT3/WNT3A DKO ones (Fig. 4f, g, top panel). Finally, to formally demonstrate that this phenotype in WNT3-deficient cells was due to the specific disruption of the gene rather than due to unspecific effects, we used exogenous WNT activation to compensate for it. Indeed, an additional treatment with CHIR99021, a GSK3 inhibitor, rescued the genetic defect to restore cardiac differentiation competence in the two WNT3-deficient lines (Fig. 4g, bottom).

## Discussion

Overall, these data suggest that the fundamental role of EOMES in germ layer induction[2,12] goes beyond activating individual factors downstream in the differentiation cascade, although the stimulation of genes like *Mesp1* is certainly very important, too[1,11]. Rather, we propose that much of the function of EOMES is intimately based on a cooperative interplay with the WNT pathway, at least in cardiac mesoderm specification. Both EOMES and WNT are crucially required for cardiac induction. Under cardiac-permissive conditions, WNT signaling activates *EOMES*[16]. Conversely, EOMES promotes WNT signaling through the induction of *WNT3* as shown here. The two entities hence form a self-sustaining regulatory module in cardiac mesoderm specification (Fig. 4h). In this model, cardiac induction may either be accomplished through exogenous WNT activation as in WNT-driven cardiac differentiation protocols[23,24], or by intrinsic *EOMES* induction as shown here—the module becomes activated in both ways. These mutual links between *EOMES* and *WNT3* are likely to have in vivo relevance as *Wnt3* is fundamentally required for mouse primitive streak formation, whereas *Wnt3a* only plays a role at later stages[25,26].

Overamplification of the module promotes non-cardiac fates. To prevent this from happening in vivo, migratory cardiac mesoderm cells assume residence at the anterior side of the embryo where they become exposed to WNT-antagonizing factors, which will inactivate the regulatory circuit and counteract anti-cardiac gene expression[16,27]. In a cell culture setting, WNT or EOMES need to be dose-controlled to prevent the module from overshooting and additionally, sustained signaling is to be inhibited in a subsequent step (Fig. 4h). In this regard, EOMES may be considered a context-dependent master regulator of cardiac specification. The fact that this key inductive function is highly dose and context-dependent may explain why it remained somewhat underestimated in a previous report using mouse ES cells[11]. Overall, the signaling context appears to have a profound impact on differentiation driven by intrinsic cell fate determinants. For instance, activation of Nodal-SMAD2/3 signaling cooperates with EOMES to redirect cell fate along the endodermal lineage[11,13]. And even classical examples such as MYOD1-driven forward programming of hESCs into skeletal muscle requires concomitant retinoic acid signaling for optimal outcomes[28]. Hence, environmental constraints do not question but may overshadow key roles of intrinsic regulators in cell fate programming.

From a practical point of view, EOMES-driven cardiac induction may, even though the approach relies on genetic modification, bear advantages for the robust production of CMs at a larger scale. pCMs display functional properties comparable to conventional hPSC-CMs but their generation could conveniently be initiated from routine cultures and appeared to be easier to control than growth factor-based differentiation in our hands. These observations hence encourage a more detailed analysis of pCM physiology to serve in drug testing and disease modeling paradigms. Furthermore, it will be interesting to investigate whether there are additional master determinants of human cardiac induction acting at more downstream stages of the differentiation process.

## Methods

**hESC culture**. WT hESCs (HuES6[29] background, obtained from Harvard University), WT hiPSCs (F1 background, fetal liver fibroblast-derived[21]), and their genetically modified derivatives were maintained on 6-well dishes coated with 1 ml/well 1:75 diluted Matrigel™ HC (Corning #354263), in defined FTDA medium[30].

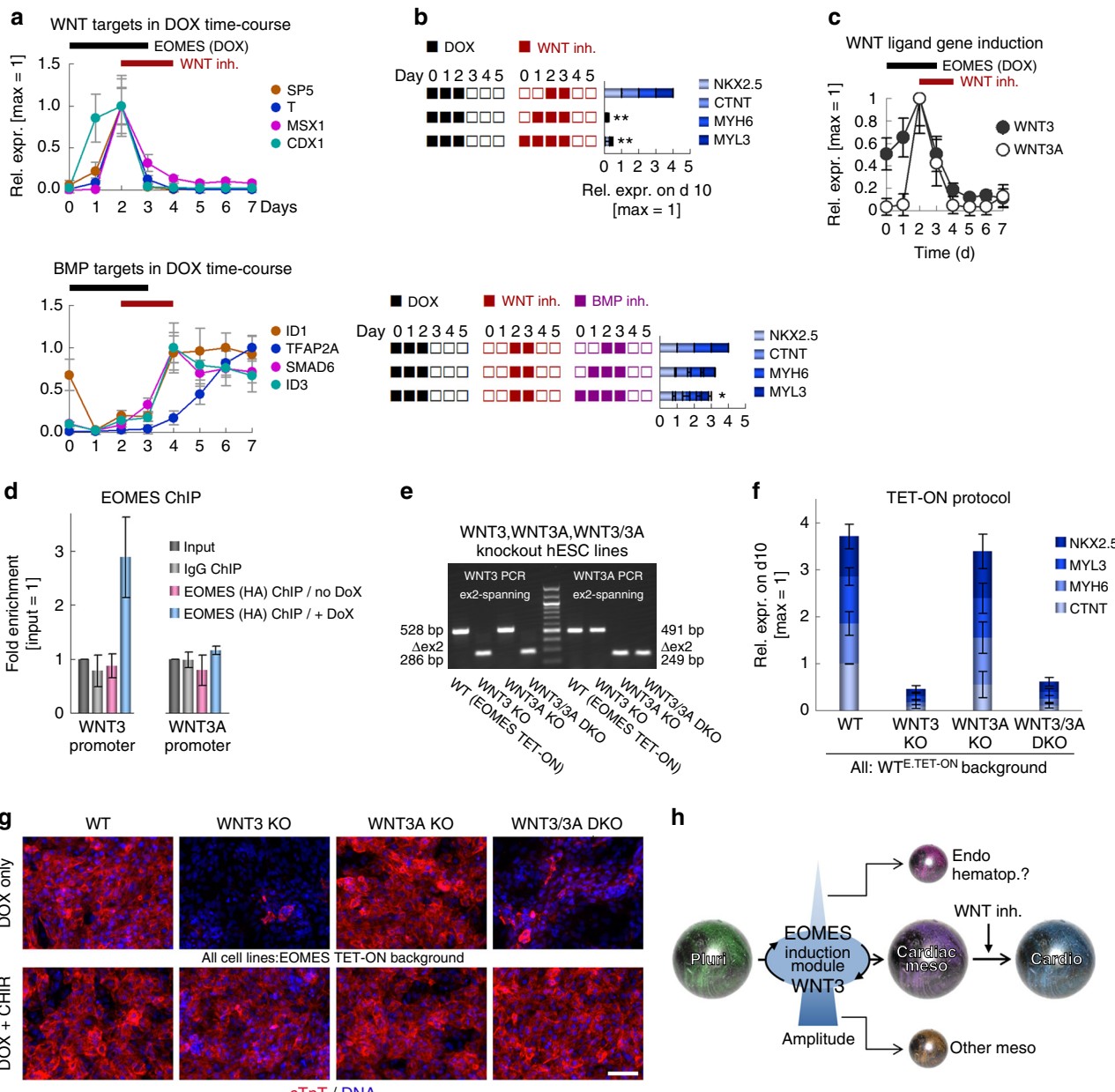

**Fig. 4** Cardiac programming by EOMES is based on the downstream induction of *WNT3*. **a** Short-term induction of canonical WNT but not BMP target genes during the time of DOX treatment (array data from Supplementary Data 2; error bars: bead s.d.). **b** Signaling inhibitor experiments with constant DOX at days 0–2 using indicated treatments with WNT (top, 200 nM C-59) and BMP (bottom, 50 nM dorsomorphin) antagonists (qPCR data, n = 3). C-59 had to be administered to all samples of the bottom experiments to provide overall CM-permissive differentiation conditions as a baseline (see Fig. 3b). Data are normalized to the top samples in each of the two panels. **c** Canonical WNT ligand gene induction in the TET-ON protocol (from Supplementary Data 2; error bars: bead s.d.). **d** EOMES ChIP-qPCR using DOX-induced hESCs differentiated until the day 2 cardiac mesoderm stage (n = 3; error bars: s.e. m.). Amplicons overlap with hit regions identified by ChIP-seq[13]. No-DOX samples served as specificity controls. **e** RT-PCR validation of frameshift-causing splice mutations induced by CRISPR/Cas9n in the *WNT3* and *WNT3A* genes (also see Supplementary Fig. 4b). All cell lines were differentiated into cardiac mesoderm prior to RNA isolation. **f** TET-ON differentiation of the indicated cell lines (n = 3; error bars: s.e.m.). **g** *WNT3* but not *WNT3A* is required for EOMES-mediated cardiac induction (top), and its disruption can be compensated by extrinsic WNT activation using CHIR99021 (8 µM, bottom panel, representative immunostains at 2 wk). Scale bar: 100 µm. **h** General model illustrating the interplay between EOMES and WNT signaling in cardiac mesoderm induction. Subsequent WNT inhibition silences the module to allow cardiac differentiation to proceed. A too weak or a too strong activation of the induction loop promotes non-cardiac fates as indicated. The regulatory link between canonical regulatory link between canonical WNT signaling and EOMES has previously been established[16]

---

FTDA was composed of DMEM/F12, 1 × PenStrep/Glutamine, 1 × defined lipids (Thermo), 1 × ITS (Corning), 0.1% human serum albumin (Biological Industries), 10 ng/ml FGF2 (PeproTech #100-18B), 0.2 ng/ml TGFβ1 (eBioscience #34-8348-82), 50 nM Dorsomorphin (Santa Cruz), and 5 ng/ml Activin A (eBioscience #34-8993-85). Fully confluent hPSC cultures were harvested by a 15–20 min incubation

with Accutase™ (Sigma) containing 10 µM ROCK inhibitor Y-27632 (abcamBio-chemicals) and seeded out for passaging into new 6-well plates at 400,000–500,000 cells per well, in FTDA + ROCKi. Cells were split every 3–4 days and kept in culture for a maximum of 30 passages. Short-term signaling stimulation experiments were carried out using semiconfluent cultures.

**Genetic manipulation.** For the induction of splice mutations, two pairs of CRISPR/Cas9 nickase vectors were designed to encompass intron–exon boundaries. CRISPR vectors were generated by oligonucleotide cloning as described[16], using the pX335 vector[31] (Addgene plasmid #42335) modified to contain a GFP-2A-puromycin selection cassette. gRNA-specific targeting sequences are given in Supplementary Table 2. For disrupting a given locus, hESCs were transfected with the four corresponding CRISPR vectors using Fugene™ HD (Promega). For simultaneously disrupting two genes, hESCs were accordingly transfected with a cocktail of eight plasmids. One day later, transfectants were enriched using transient puromycin selection for 1 d (0.5 μg/ml). Two days later, semiconfluent cultures were replated at clonal density. Half the cells from single emerging colonies were used for DNA isolation and gPCR screening using deletion-spanning primers given in Supplementary Table 2. PCR-positive candidate clones vs. total number of clones screened: 3/24 (WNT3), 4/32 (WNT3A), 1/24 (WNT3/WNT3A DKO). The remaining half-colonies from positive clones were expanded and (re-)validated by diagnostic PCRs on gDNA and cDNA templates, as well as by sequencing of multiple TOPO/TA clones. One cell clone per genotype was used in experiments.

Clonal DOX-inducible overexpression lines were generated using PiggyBac transposition as described[16]. Three vectors containing PiggyBac-flanked inducible EOMES-HA-IRES-Venus, PiggyBac-flanked constitutive rtTA-IRES-NEO, and constitutive transposase transgenes were co-transfected into hPSCs using Fugene HD, at a DNA mass ratio of 10:1:3, respectively. Stable transgene-positive cells were selected using 50 μg/ml G418 and replated at low density. Emerging colonies were split in half and replated. One plate was test-induced for 1 day using DOX, and if homogeneous green fluorescence was observed for a given clone the uninduced replicate culture was picked, expanded, and characterized as appropriate.

**Differentiation.** Standard growth factor-mediated cardiac induction was performed under serum and albumin-free conditions as described[17]. Briefly, fully confluent hESC cultures were harvested using Accutase resuspended in day 0 differentiation medium and seeded out at 500,000 cells/well in Matrigel-coated 24-well plates (2 ml volume/well). Day 0 differentiation medium consisted of KO-DMEM, 1 × ITS, 10 μM Y-27632, 1 × PenStrep/Glutamine, 10–20 ng/ml FGF2, 0 or 5 ng/ml Activin A, 0.5–1 ng/ml BMP4 (R&D #314-BP-050), and 1 μM CHIR99021 (AxonMedchem #Axon 1386). From day 1 onward, the basal differentiation medium consisted of KO-DMEM, 1 × TS (transferrin/selenium), 250 μM 2-phospho-ascorbate, and PenStrep/Glutamine. WNT inhibitor C-59 (Tocris #5148) was added to the cultures from 48 to 96 h of differentiation at 0.2 μM. Optionally, for promoting an atrial fate, all-trans-retinoic acid (Sigma #R2625) was supplemented from 72 to 120 h of differentiation. Differentiation medium was changed on a daily basis.

For investigating non-cardiac differentiation fates, the above culture conditions were modified according to a previously established mesendodermal patterning model[16], by varying the initial BMP4 dose administered, or by additional Activin A supplementation, and/or by omitting the addition of the WNT inhibitor in the second differentiation step. Neural differentiation was promoted by standard embryoid body-based procedures.

For EOMES-driven cardiac differentiation, transgenic hESCs were either replated into Matrigel-coated 24-well plates in FTDA + ROCKi (1 M cells/well), to initiate differentiation the day after or, alternatively, differentiated directly from subconfluent (day 3) maintenance cultures in 12 or 6-well format. Basal differentiation medium was identical to that in the growth factor-based protocol. Insulin addition on day 0 was optional. DOX was administered for the first 3 days at 0.25 μg/ml in case of the two HuES6 derivative cell lines, or at 0.1–0.13 μg/ml in case of the WT$^{E.TET-ON}$ hiPSC line. C-59 was applied from day 2 to 3 at 0.2 μM unless stated otherwise. For investigating non-cardiac differentiation with the TET-ON protocol using WT$^{E.TET-ON}$ hESCs, the DOX concentration used was lowered (0.125 μg/ml) or increased (1 μg/ml). Cultures differentiated this way were analyzed by means of microarrays followed by validation of selected markers at the protein level.

For CM maturation, beating monolayers emerging at day 6–8 were dissociated using TrypLE Select (Thermo) on day 8–10 and replated at a ratio of ~1:4 using CM splitting medium which consisted of RPMI 1640 (Thermo), 1 × ITS, 0.1% HSA, 250 μM phospho-ascorbate, 0.008% thioglycerol, 1 × PenStrep/Glutamine, and 10 μM ROCKi. Next day, medium was replaced by CM maintenance medium consisting of KO-DMEM, 1 × ITS, 0.1% (w/v) HSA, 1 × defined lipids, 250 μM phospho-ascorbate, 0.008% thioglycerol, and PenStrep/Glutamine. 1.5–2 wk later, cultured CMs were used for downstream analyses as indicated.

**RT-qPCR.** RNA was isolated using NucleoSpin RNA kits with on-column DNase treatment (Machery Nagel). Reverse transcription was performed using M-MLV reverse transcriptase (Affymetrix #78306) with oligo-dT$_{15}$ priming at 42 °C. Real-time PCR was carried out using validated primers given in Supplementary Table 2 and BioRad iTaq™ Universal SYBR Green Supermix on ABI instrumentation. The ΔΔCt method was used to calculate relative transcript abundance against a fixed reference sample or against the highest expressing sample of a given series. Alternatively, results were expressed relative to a housekeeping gene standard (RPL37A, $2^{-\Delta Ct}$). Statistics were based on RPL37A-corrected Ct values or fold expression changes, as appropriate. Conventional RT-PCRs were performed according to standard procedures.

**Genome-wide expression analysis.** Labeled cRNA was prepared from 500 ng DNA-free RNA samples using TotalPrep™ linear RNA amplification kits (Thermo #AMIL1791). Microarray hybridizations on Illumina V4 human HT-12 bead arrays were carried out as recommended by the manufacturer. Cy3-stained chips were scanned using HiScan SQ instrumentation. Background subtraction and cubic spline normalization was done using GenomeStudio software. Processed data were filtered in MS Excel by setting experience-based thresholds for expression changes and minimal gene expression levels. For functional annotation of filtered gene lists, employing the Ensembl BioMart interface, array probe sequences were converted into GRCh37/hg19 genome coordinates which were then used as input for GREAT analysis[32]. Statistically significant hits were subjectively filtered for biological relevance and presented based on the obtained p-values.

**ChIP-qPCR.** Chromatin immunoprecipitation was performed as described[16], using samples differentiated until the cardiac mesoderm stage (day 2). HA tag-based ChIP (Santa Cruz #sc-805-X) was used due to the temporal unavailability of a ChIP-grade EOMES antibody. IgG control was Santa Cruz #sc-2027. qPCRs were performed using diluted ChIP and serial dilutions of input DNA. Fold enrichments over input were calculated following internal normalization to an irrelevant control locus. Primers are given in Supplementary Table 2.

**Immunocytochemistry.** Immunofluorescence analysis was carried out according to standard procedures using secondary Alexa-488 or Alexa-568-conjugated antibodies and Hoechst for fluorescent staining. Primary antibodies were α-actinin (Sigma #A7811, 1:800), albumin (R&D #MAB1455, 1:100, albumin-free staining procedure), ANP (R&D #AF3366, 1:100), α-fetoprotein (Sigma A8452, 1:500), COL3A1 (Santa Cruz #sc-8780-R, 1:100), HA tag (Santa Cruz #sc-806-X, 1:200), MLC2v (ProteinTech Group #10906-1-AP, 1:200), NKX2.5 (R&D #AF2444, 1:100), PECAM1 (R&D #BBA7, 1:50), smooth muscle actin (DakoCytomation #M085129-2, 1:100), SOX2 (R&D #AF2018, 1:200), cardiac troponin I (cTnI, Santa Cruz #sc-15368, 1:200), cardiac troponin T (cTnT, Thermo #MS-295-P, 1:200), renin (R&D #AF4090, 1:150), β-III-tubulin (Covance #MMS-435P, 1:500), and vimentin (Sigma #V6630, 1:200).

**Immunoblotting.** Western blotting was performed according to standard procedures using peroxidase-conjugated secondary antibodies and SuperSignal® West Pico chemiluminescent substrate (Thermo). Primary antibodies were EOMES (Abcam #ab23345, 1:1000), GAPDH (Thermo #AM4300, 1:10,000), and NKX2.5 (R&D #AF2444, 1:100). All uncropped immunoblots can be found in Supplementary Fig. 5.

**Flow cytometry.** Flow cytometry was carried out following Accutase digestion of primary monolayers or replated CMs, fixation with 2% formaldehyde, and blocking/antibody incubations in FACS buffer (0.5% saponin/5% fetal calf serum in PBS). Antibodies used were cTnT (Thermo #MS-295-P, 1:200) and Alexa-488 anti-mouse (Thermo #A11001, 1:1000).

**Electrophysiological analysis.** For analysis of pCMs on microelectrode arrays (USB-MEA256 system, Multichannel Systems), the electrode areas of plasma-cleaned 9-well MEAs were coated with 3 μl of a 1:75 diluted Matrigel solution in KO-DMEM for approximately 2 h at 37 °C in a humidified cell culture incubator. pCMs were dissociated from maintenance cultures using a 10 × TrypLE Select digestion to obtain a single-cell/small aggregate suspension. Coating solution was removed from the electrode arrays to be replaced by 25,000–50,000 cells resuspended in a ~3 μl droplet of CM splitting medium. CMs were allowed to attach for ~30 min. Subsequently, MEA chambers with attached cells were filled with 150 μl of CM replating medium. Next day, medium was changed to CM maintenance medium. From the following day onward, cell preparations were used for recordings at 37 °C. Drugs were washed in for 5 min (isoprenaline: 10 μM, propranolol: 10 μM, E-4031: 100 nM, 4-AP: 1 mM). Algorithms for determining field potential durations as QT$_{max}$-like intervals were implemented in MC Rack software v4.5.7. Extracted values were averaged from independent replicates. Beat frequency correction was based on Bazett's formula.

**Statistics and reproducibility.** Essentially every result was confirmed in at least one additional experiment. The TET-ON protocol has independently been reproduced multiple times by three individuals, with highly similar outcomes. Unless stated otherwise, numerical data represent means of n biological replicates, as indicated in figure legends. Error bars of qPCR data denote s.e.m., and those of microarray data reflect bead standard deviation. Where meaningful, 1 or 2-sided unpaired t-tests were used, as appropriate, to compare pairs of samples in MS Excel. Asterisk denotes a significance level of $p < 0.05$, and **$p < 0.01$. Statistics on microarray data were based on an Illumina custom model implemented in GenomeStudio software.

**Data availability.** The authors declare that all data supporting the findings of this study are available within the article and its Supplementary Information files or from the corresponding author upon reasonable request. Raw and processed

microarray data have been deposited in the Gene Expression Omnibus (NCBI GEO) database under accession codes GSE97627 (cardiac induction time courses related to Supplementary Data 2) and GSE97625 (short-term stimulation experiment related to Supplementary Data 3).

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

## Acknowledgements

This work was supported by the Chemical Genomics Centre of the Max Planck Society. I.P. acknowledges grant support by the IMF (University of Münster Medical School).

## Author contributions

M.J.P., R.Q., I.P., J.F., J.R. and B.G. designed and performed the experiments, and analyzed the data. A.R. performed karyotyping. G.S. and B.G. provided financial support. B. G. wrote the manuscript with input from co-authors.

## Additional information

**Competing interests:** A patent application has been filed based on findings in this study, with B.G. and M.J.P. as inventors: "Facilitated generation of cardiomyocytes by forward programming of human pluripotent stem cells" (EP17186131, 8/2017). The remaining authors declare no competing financial interests.

