## [Peer Review File · Nature Communications]

Reviewers' Comments:

Reviewer #1:

Remarks to the Author:

GENERAL COMMENTS

Pfieffer and colleagues present a manuscript examining the role of eomesodermin (EOMES) in regulating cardiac mesoderm induction using a human ESC model. This transcription factor is known to be essential in germ layer development particularly of endoderm and mesoderm based on prior studies. In this manuscript, the authors generate a knockout of EOMES in human ESCs and show that this clearly blocks cardiac differentiation using a standardized protocol that can be rescued with a genetically engineered EOMES dox-inducible construct. Most importantly, the authors implicate the role of EOMES in inducing Wnt3 expression as the key downstream effector that promotes the formation of cardiac mesoderm using additional chemical inhibition and knockout lines. The authors conclude with a model of a feedback cardiac induction module of EOMES-Wnt. Overall, the manuscript is clearly written and presents novel findings regarding the impact of EOMES on Wnt3 expression. However, there are some significant limitations as currently presented as follow:

1. A single human ESC lines was apparently used in this study, HuES6, and it is important to confirm that the findings are generalizable and not unique to this stem cell line. For example, given the known variations in paracrine signaling in different stem cell lines, different results could be observed in different cell lines (e.g. Kattmann et al. 2011, Cell Stem Cell, 8(2):228-40 for individual cell line titration of growth factors in cardiac differentiation). Repeating a key experiment or confirmatory experiment in another cell line would strengthen the manuscript.
2. The authors' feedback model of EOMES and Wnt3 signaling is oversimplified. While the authors provide clear evidence that EOMES can stimulate Wnt3 expression and signaling, the converse is not clear from the presented data – that Wnt3 can stimulate EOMES expression or signaling. The effect of Wnt signaling is context-dependent, so Wnt effects on cardiac progenitors are likely quite different than on induction of cardiac mesoderm. Either the authors need to revise their model or provide such evidence.

MINOR COMMENTS

1. Should 'eomesodermin' be used in title rather than EOMES for clarity?
2. On page 3, the authors state that "To verify this conclusion in a more functional way, pCMs were treated with E-4031, an inhibitor of the hERG potassium channel that plays a prominent role in ventricular repolarization." Although it is true that hERG channels contribute to ventricular repolarization, they also contribute to atrial repolarization prominently and so do not discriminate between cardiomyocytes from these chambers.
3. On page 3, "...and slowdown after addition of propranolol which is a beta blocker (Fig. 2d)." Why should a beta blocker slow the endogenous rate in the absence of adrenergic stimulation or innervation?
4. On page 4, "...easily be overdosed." Overdose is the wrong word here as doses are given to patients or animals.
5. Suppl Fig 2c – Y axis label of QTc interval needs to be more clearly labeled as it is not true QTc which comes from surface electrocardiograms. Some indication that this is not standard QTc such as QTc-like or Pseudo-QTc, etc should be used.
6. Reference 8 is incomplete.

Reviewer #2:

Remarks to the Author:

This paper investigates EOMES during cardiac differentiation of human embryonic stem cells (hESCs). The finding that EOMES is essential for cardiac differentiation (Fig. 1) has been published previously by this group (Rao et al Cell Stem Cell 2016).

Eomes is a key driver of mesoderm formation, including the driving epithelial-to-mesenchymal transitions necessary for nascent mesoderm formation. Thus, it is highly likely that a failure to form mesoderm underlies the phenotype rather than specifically impaired cardiomyocyte differentiation. In this context, the authors should examine the capacity of the EOMES knockout to form other mesodermal lineages. The data establish the endoderm differentiation is compromised in EOMES knockouts. Furthermore, Tet-ON EOMES patterned day 2 differentiated cells should be examined for the capacity to produce other mesodermal lineages (eg. blood) and endodermal derivatives.

Similarly, overexpression of EOMES using PiggyBac vectors promotes mesodermal markers (Fig. 2). The cardiogenic potential of this mesoderm can be exploited by Wnt inhibition, as has been established. However, whether mesoderm derived by the TET-ON EOMES over expression is specific to the cardiac lineage has not been evaluated. The authors should test if EOMES induced mesoderm retains the capacity to form other mesodermal derivatives.

The finding that EOMES can substitute for cytokine/small molecules for mesoderm induction is an interesting one and may be of utility for biotechnology approaches. However, at present the expression is based on random integration of the PiggyBac vectors. Have the authors used a single integration site such as AAVS1 to control for possible integration defects (eg. silencing, mutagenesis)?

The major finding is establishing a link between EOMES and WNT3 signalling during early mesoderm specification. Inhibiting WNT signalling or deleting WNT3 prevents the development of cardiomyocytes, presumably because mesoderm does not form. Indeed, Wnt3 knockout mice do not form a primitive streak. No data is shown when WNT3 is deleted in a wild type background, the hypothesis would be that WNT3 is necessary for mesoderm formation. This control is necessary to further establish that EOMES drives mesoderm development via a WNT3-dependant mechanism. Furthermore, WNT and BMP signalling act synergistically during mesoderm formation and BMP4 is unregulated by EOMES (Table S2). While a BMP inhibition experiment is performed (Fig. 3d) it is done in the context of WNT inhibition. Have the authors tested BMP inhibition alone?

Another concern is the use of only a single hESC line. The key findings need to be confirmed in another genetic background to ensure that these findings are not restricted to the HuES6 cell line.

Minor points

Regarding "the issue of whether there is a bona fide master regulatory factor specifically promoting the induction of cardiac cells", multiple lines of evidence suggest that there is no single key master cardiogenic factor. This is due to the nature of the highly conserved gene regulatory network that controls cardiogenesis. While removal of a single member of this network often results in lethality in all cases contractile cardiomyocytes are formed. Furthermore, attempts to make induced cardiomyocytes have shown that at least 3 transcription factors (GATA4, TBX5, MEF2c) are required to reprogram fibroblasts. Thus, a "master regulatory" factor for cardiomyocytes does not exist. This should be clarified in the introduction.

Please separate the genes in the gene expression graphs (eg Fig 1 , Fig. 3 d, e). Combining multiple genes in one bar makes it difficult to interpret the data. This is used throughout the manuscript. Also, when relative gene expression is used please indicate in the figure legend what

the expression is relative to (eg. relative to wildtype etc).

Have the authors examined the onset of mesodermal markers (CD13, ROR2, PDGFRa etc) in differentiating EOMES knockout cell lines? The microarray data (Table S1) shows that transcripts for PDGFRa are up-regulated in the TET-On EOMES samples whereas ROR2 levels do not vary. These data may be useful in quantifying mesodermal cell production in the various cell lines used.

How many PiggyBac lines are used in the experiments? Is all the data from a single clone or are multiple lines used to confirm the findings?

Figure 2 legend: “~1.5 weeks” and “~3 weeks” are somewhat vague. Please provide exact days of differentiation.

The modified hESC lines were not karyotyped to ensure that no abnormalities arose during the genetic manipulation and no evidence is shown that they maintain pluripotency markers.

Figure 3 please provide a schematic of the WNT3 and WNT3a alleles showing the region deleted and the sequence data demonstrating that the allele. Also, a second PCR should be performed using a primer internal to the CRISPR mediated deletion to confirm the knockout. The smaller PCR product may preferentially amplify meaning that a heterozygous genotype may be misclassified as a homozygote. An internal/external PCR combination should only produce a product if a wild type allele is still present. This could be presented in supplementary data.

Figure 3d please indicate on the figure that the WNT inhib. was included with the BMP inhibitor. This experiment should also be repeated using the BMP inhibitor alone.

The microarray data presented in Table S1 could be presented more effectively. A summary of the data for each transcript (rather than each spot on the microarray), including a column showing fold differences between the samples, should be included as a separate table within the excel spreadsheet. This would make it easier for the reader to identify which gene networks are dysregulated.

The model in Fig. 3j suggests the EOMES-WNT form a cardiac mesoderm induction module, however, further data are required to confirm that this indeed cardiac specific mesoderm. Further differentiation of these cells along blood, endothelial and other mesodermal lineages is required to formally establish that EOMES-WNT specific pre-cardiac mesoderm.

Reviewer #3:

Remarks to the Author:

The role of Eomes in cell fate specification and in regulating WNT3 expression is novel and important. I find this elegant study very interesting and overall convincing and highly recommend publication after addressing the following points:

1. The treatment of WNT3 KO cells with CHIR compound in Fig.3i is a nice specificity control, but a more quantitative readout of the rescue of cardiac differentiation would strengthen the conclusion. In addition, Methods should state how many independent CRISPR clones were analyzed per genotype.
2. Please indicate the statistical significance of the GO term enrichments shown in Fig. 1d.
3. The legend of Suppl. Fig. 2a states that n was 2 to 5. To estimate the significance of the

cumulative effect on cardiac fate, the authors should state which differences did or did not reach statistical significance.

4. In Suppl. Fig. 2c, 'time ---- 50 ms' probably denotes the scale of the x-axis, but this should be clarified (ideally by drawing an x-axis with ticks).

5. A description of Fig. 3e seems to be missing in the text.

We wish to thank all reviewers for expressing interest in our work and evaluating it in a fair and constructive manner. We do believe that the various changes to the manuscript (highlighted in blue font color) and the additional data provided in this revised version will further strengthen our conclusions and increase the overall value of this work.

Reviewer #1 (Remarks to the Author):

GENERAL COMMENTS

Pfieber and colleagues present a manuscript examining the role of eomesodermin (EOMES) in regulating cardiac mesoderm induction using a human ESC model. This transcription factor is known to be essential in germ layer development particularly of endoderm and mesoderm based on prior studies. In this manuscript, the authors generate a knockout of EOMES in human ESCs and show that this clearly blocks cardiac differentiation using a standardized protocol that can be rescued with a genetically engineered EOMES dox-inducible construct. Most importantly, the authors implicate the role of EOMES in inducing Wnt3 expression as the key downstream effector that promotes the formation of cardiac mesoderm using additional chemical inhibition and knockout lines. The authors conclude with a model of a feedback cardiac induction module of EOMES-Wnt. Overall, the manuscript is clearly written and presents novel findings regarding the impact of EOMES on Wnt3 expression. However, there are some significant limitations as currently presented as follow:

1. A single human ESC lines was apparently used in this study, HuES6, and it is important to confirm that the findings are generalizable and not unique to this stem cell line. For example, given the known variations in paracrine signaling in different stem cell lines, different results could be observed in different cell lines (e.g. Kattmann et al. 2011, Cell Stem Cell, 8(2):228-40 for individual cell line titration of growth factors in cardiac differentiation). Repeating a key experiment or confirmatory experiment in another cell line would strengthen the manuscript.

Although we had already demonstrated pCM formation in two clonal TET-ON cell lines in the first version of the manuscript, these admittedly share the same genetic background. We have hence prepared an additional cell line, on induced pluripotent stem cell background, which involved the screening of ~50 candidate clones to identify at least one that would homogeneously express EOMES in a DOX concentration-dependent manner, as required. Validation data of one selected line are now shown in Fig. S3_{e,f}, and highly efficient cardiomyocyte programming is demonstrated in Fig. 3_c. Hence, these data suggest that EOMES universally induces a cardiac mesoderm fate both in hES and hiPS cells. We thank this reviewer for encouraging these significant but useful efforts.

2. The authors' feedback model of EOMES and Wnt3 signaling is oversimplified. While the authors provide clear evidence that EOMES can stimulate Wnt3 expression and signaling, the converse is not clear from the presented data – that Wnt3 can stimulate EOMES expression or signaling. The effect of Wnt signaling is context-dependent, so Wnt effects on cardiac progenitors are likely quite different than on induction of cardiac mesoderm. Either the authors need to revise their model or provide such evidence.

We do not deny that the model presents a simplified view on the complex process of cardiac induction, as there are certainly many more genes involved downstream of the module. We do think, though, that it brings the key findings of our work across in a crisp and intuitive manner. One aspect that was admittedly not highlighted in the previous version of the model is the dose-dependence. Through additional experimentation, we have now investigated differentiation fates promoted outside the DOX concentration range that is most optimal for cardiac induction. These additional and revealing findings are shown in revised Fig. S3_{c,d} and they have also been incorporated into the model of Fig. 4_h - together with indicating the overall dose-dependence of the system, in agreement with developmental principles.

As to the link between WNT and EOMES: In the legend to Fig. 4_h, we are now referring to our previous work that the present study is in part based on (Rao et al., 2016, Cell Stem Cell 18:141-53). In that paper, we show that EOMES activation by canonical WNT signaling is a first key event of cardiac mesoderm induction. Indeed, some contemporary cardiac differentiation protocols are exclusively based on canonical WNT signaling activation. We apologize for not having stressed out the link to our previous study in a sufficiently clear manner.

MINOR COMMENTS

1. Should 'eomesodermin' be used in title rather than EOMES for clarity?

We intentionally use "EOMES" instead of "eomesodermin" throughout, in part to better highlight the genetic nature of our new approach. Since "eomesodermin" is another commonly used designation, though, we acknowledge the recommendation and now mention this name in the revised abstract.

2. On page 3, the authors state that "To verify this conclusion in a more functional way, pCMs were treated with E-4031, an inhibitor of the hERG potassium channel that plays a prominent role in ventricular repolarization." Although it is true that hERG channels contribute to ventricular repolarization, they also contribute to atrial repolarization prominently and so do not discriminate between cardiomyocytes from these chambers.

The claim that hERG would not significantly contribute to atrial repolarization was based on some of our recent observations in the context of cardiac subtype-specific disease modeling in the hPSC system (Marczenke et al., PMID 28729840). Admittedly, however, this may not present an accepted view and it might overall be confined to the hPSC-CM system. And in general, it is totally true that there are no strict ventricular-specific potassium channels. We have therefore decided to merely mention the result as a selected demonstration of pan-cardiac ion channel functionality (page 3 of revised manuscript). Instead, in Fig. S2_{r,d}, we are now showing new data generated using 4-AP as an atrial-specific ion channel inhibitor. We hope RI agrees that these physiological data, together with the previous gene expression analysis, now support our conclusion - that pCM are ventricular-like by default - in a more convincing way.

3. On page 3, "...and slowdown after addition of propranolol which is a beta blocker (Fig. 2d)." Why should a beta blocker slow the endogenous rate in the absence of adrenergic stimulation or innervation?

We consistently observe a decrease in beating rates using beta blocker treatment of hPSC-CMs, which is thus not specific to pCMs (see e.g. Fig. S4 in Marczenke et al., PMID 28729840). This would simply suggest that there is a certain baseline cAMP pathway activity in the cells.

4. On page 4, "...easily be overdosed." Overdose is the wrong word here as doses are given to patients or animals.

We have followed the recommendation and replaced the phrase by "under or overstimulated" (page 4 of revised ms).

5. Suppl Fig 2c – Y axis label of QT_c interval needs to be more clearly labeled as it is not true QT_c which comes from surface electrocardiograms. Some indication that this is not standard QT_c such as QT_c-like or Pseudo-QT_c, etc should be used.

We agree. In new Fig. S2_{r,d}, as well as in the text throughout, we now speak of field potential durations or "QT_{max}-like" intervals.

6. Reference 8 is incomplete.

This has been corrected (thanks a lot for taking such a close look).

Reviewer #2 (Remarks to the Author):

This paper investigates EOMES during cardiac differentiation of human embryonic stem cells (hESCs). The finding that EOMES is essential for cardiac differentiation (Fig. 1) has been published previously by this group (Rao et al Cell Stem Cell 2016).

Eomes is a key driver of mesoderm formation, including the driving epithelial-to-mesenchymal transitions necessary for nascent mesoderm formation. Thus, it is highly likely that a failure to form mesoderm underlies the phenotype rather than specifically impaired cardiomyocyte differentiation. In this context, the authors should examine the capacity of the EOMES knockout to form other mesodermal lineages. The data establish the endoderm differentiation is compromised in EOMES knockouts. Furthermore, Tet-ON EOMES patterned day 2 differentiated cells should be examined for the capacity to produce other mesodermal lineages (eg. blood) and endodermal derivatives. Similarly, overexpression of EOMES using PiggyBac vectors promotes mesodermal markers (Fig. 2). The cardiogenic potential of this mesoderm can be exploited by Wnt inhibition, as has been established. However, whether mesoderm derived by the TET-ON EOMES over expression is specific to the cardiac lineage has not been evaluated. The authors should test if EOMES induced mesoderm retains the capacity to form other mesodermal derivatives.

These are interesting and insightful points brought up by R2. Although we felt that delving into alternative non-cardiac differentiation protocols would go beyond the scope of our study, we found it useful to investigate alternative fates based on our present differentiation platform using moderately varied induction conditions. Indeed, we had previously demonstrated that terminal differentiation outcomes on our platform are highly dependent on the concentrations of the inductive signaling cues, as well as on subsequent WNT inhibition (Rao et al., 2016, Cell Stem Cell 18:141-53). We have therefore approached the questions from two angles:

In a first set of additional experiments, we investigated non-cardiac mesoderm differentiation competence in EOMES KO cells. This is described in results and methods (page 2/3 and 7 of the revised manuscript, respectively) and a corresponding unbiased expression analysis is shown in Fig. 1,d. Confirmatory immunocytochemical stainings are shown in new Fig. S1_r. Although we did not specifically monitor blood differentiation markers, these data interestingly reveal that the EOMES knockout preserves differentiation competence into several mesodermal cell types except cardiomyocytes.

In a second experimental series, we followed up on the point that cardiac induction was strictly dose-dependent, a fact that is now also reflected in the revised model of Fig. 4,h and that we could now also demonstrate using an independent TET-ON line based on induced pluripotent stem cells (Fig. 3_rc and Fig. S3_re,f). Outside the pro-cardiac range, low or high-DOX induction alternatively gave rise to hepatic, putative hematopoietic, smooth muscle, as well as mesenchymal differentiation fates (see results text on page 4, data in Fig. S3_rc,d, and extended model in Fig. 4_rh).

We strongly hope these efforts adequately address the points by R2 in the context of this overall cardiac-centred work. Certainly, non-cardiac mesoderm fates could be further investigated in dedicated follow-up studies and we will be happy to share our cell lines with the community for this purpose.

The finding that EOMES can substitute for cytokine/small molecules for mesoderm induction is an interesting one and may be of utility for biotechnology approaches. However, at present the expression is based on random integration of the PiggyBac vectors. Have the authors used a single integration site such as AAVS1 to control for possible integration defects (eg. silencing, mutagenesis)?

We have not used safe-harbour integration of the expression cassette. It is true that with random integration, silencing of the transgenic cassette can in principle occur. For this reason, established candidate lines need to be carefully validated. For example, upon establishing our new TET-ON hiPSC line in course of the present revision, we have screened approximately 50 candidate clones for an efficient interplay between the TET-ON cassette and the transactivator, as well as for homogeneous transgene induction over at least several days of course. Once established, though, a good cell line tends to efficiently induce the transgene even after many passages, so it is a reliable and powerful system.

Our findings are highly unlikely to result from unspecific effects. Firstly, we are investigating a clearcut gain-of-function approach by which the previously established EOMES knockout could be faithfully rescued. Secondly, we show that this

pro-cardiac phenotype is strictly dependent on dose-dependent transgene induction, while confirming in Fig. S1_r and S3_r c that the cells are also capable of forming alternative cell fates. Thirdly, we delineate a specific molecular mechanism of EOMES action that is dependent of WNT3 activation and that could be abolished using WNT3 disruption and in turn be rescued using exogenous WNT activation. And finally, we demonstrate highly similar pCM formation outcomes in three independent cells on two different genetic backgrounds. We think these facts collectively exclude the possibility of unspecific mutagenic effects.

The major finding is establishing a link between EOMES and WNT3 signalling during early mesoderm specification. Inhibiting WNT signalling or deleting WNT3 prevents the development of cardiomyocytes, presumably because mesoderm does not form. Indeed, Wnt3 knockout mice do not form a primitive streak. No data is shown when WNT3 is deleted in a wild type background, the hypothesis would be that WNT3 is necessary for mesoderm formation. This control is necessary to further establish that EOMES drives mesoderm development via a WNT3-dependant mechanism. Furthermore, WNT and BMP signalling act synergistically during mesoderm formation and BMP4 is unregulated by EOMES (Table S2). While a BMP inhibition experiment is performed (Fig. 3d) it is done in the context of WNT inhibition. Have the authors tested BMP inhibition alone?

In our mechanistic analysis, we were particularly interested in elucidating a mechanism by which EOMES drives the process in our TET-ON system. We are not saying this is the only mechanism but we show that the activation of autocrine WNT signaling downstream of EOMES is essential. In the context of mouse in vivo studies alluded to by R2, we think the close interplay between EOMES and WNT revealed here and in our prior study (Rao et al., 2016, Cell Stem Cell 18:141-53) makes it more understandable that both EOMES and WNT3 are such important players, as discussed in the manuscript and indicated in the model.

We are not actually sure whether WNT3 disruption would render signaling factor-driven cardiac differentiation impossible - probably not, because in the standard protocol, WNT is activated by the extrinsic supplementation of CHIR. Indeed, we show that CHIR addition can rescue WNT3 deficiency in the context of the TET-ON approach (Fig. 4_r,g, bottom). Hence, it is likely that e.g. elevating the initial CHIR dose applied to the cells could readily overcome a genetic defect in WNT3 using the signaling factor-based differentiation protocol. In any case, the WNT3 KO phenotype in such a setting would not question our findings about EOMES' mode of action and therefore, we did not pursue knocking out the WNT3 gene in wild-type cells.

As to the role of BMP signaling, we fully acknowledge its contribution in the standard protocol as we have ourselves demonstrated a close collaboration with WNT signaling in the aforementioned Rao et al. study. We therefore considered both pathways as candidates for being involved downstream of EOMES, but our unbiased analysis of WNT and BMP target gene activation together with the pathway inhibition experiments in Fig. 4_r,a,b just did not reveal any substantial contribution by BMP. In course of this revision, we have carried out the suggested BMP inhibition experiment in the absence of the WNT inhibitor. As the below data shows, however, the WNT inhibition step is indeed crucial for enabling overall cardiomyocyte differentiation as a baseline:

For clarification, we are now stating this fact in the legend to Fig. 4_r,b while referring to existing data in Fig. 3_r,b: "C-59 had to be administered to all samples of the bottom experiments to provide overall CM-permissive differentiation conditions as a baseline (see Fig. 3b)." In sum, BMP signaling is a weaker activator of EOMES than WNT (Table S2 and Rao et al.) and conversely, the EOMES TET-ON approach involves an immediate activation of autocrine WNT rather than BMP signaling.

Another concern is the use of only a single hESC line. The key findings need to be confirmed in another genetic background to ensure that these findings are not restricted to the HuES6 cell line.

We agree that showing EOMES-driven cardiac induction in fully independent cell lines is key to demonstrating a universal principle. As already mentioned, therefore, we have prepared such a cell line on an induced pluripotent stem cell background and utilize it to demonstrate DOX dose-dependent cardiac induction. The corresponding data is in Fig. 3,c and S3,e,f (also see results text on page 4 and added information in methods).

Minor points

Regarding “the issue of whether there is a bona fide master regulatory factor specifically promoting the induction of cardiac cells”, multiple lines of evidence suggest that there is no single key master cardiogenic factor. This is due to the nature of the highly conserved gene regulatory network that controls cardiogenesis. While removal of a single member of this network often results in lethality in all cases contractile cardiomyocytes are formed. Furthermore, attempts to make induced cardiomyocytes have shown that at least 3 transcription factors (GATA4, TBX5, MEF2c) are required to reprogram fibroblasts. Thus, a “master regulatory” factor for cardiomyocytes does not exist. This should be clarified in the introduction.

We wish to mention that MESP1 had previously - and perhaps somewhat prematurely - been termed a cardiac master regulator based on ES cell studies which we refer to in the introduction. Furthermore, we do not make any claims about trans-reprogramming of e.g. fibroblasts into cardiac cells. It is in fact likely that this will not work using EOMES induction alone. Instead, we term it a "context-dependent master regulator", which does not present an overstatement we think. As explained in the discussion section, this activity requires the pluripotent stem cell context, the ability to tightly control EOMES expression levels, and to inhibit endogenous WNT signaling in a subsequent step. Under these conditions, however, EOMES drives near-homogeneous cardiac induction, which is pretty remarkable we think and this should also be reflected in a strong metaphoric designation in our opinion.

Please separate the genes in the gene expression graphs (eg Fig 1 , Fig. 3 d, e). Combining multiple genes in one bar makes it difficult to interpret the data. This is used throughout the manuscript. Also, when relative gene expression is used please indicate in the figure legend what the expression is relative to (eg. relative to wildtype etc).

It is overall useful to monitor more than just on marker gene in a given context, as this tends to generate more robust data. On the other hand, we think it is important to present the data in a manner as intuitive and simple as possible. At least this is what we attempt to do. Piling up genes in bar charts is one very useful measure to achieve a good compromise in these regards we think. We therefore wish to hold on to this way of presenting our data, also because even detailed information can be extracted from it. For example, the CM characterization data of Fig. S2,c would otherwise look unnecessarily complicated and consume far too much space.

qPCR data can be expressed in different ways and it is legitimate to be somewhat flexible with this in order to bring the essence of the data across in the most intuitive manner. When plotting time-course expression data of several genes into one chart, for instance, it can be most useful to normalise the data to the highest expressing sample - to best convey the respective temporal expression patterns (rather than highlighting more "absolute" expression levels for instance). Likewise, normalising the expression of differentiation markers against undifferentiated cells - which essentially do not express those genes - can be problematic due to rather strong baseline fluctuations in the ground state. For clarification, therefore, we have now stated more clearly in the methods section which ways of expressing qPCR data have overall been used in this study (page 8). Moreover, the specific normalization procedure employed is also indicated on the y axis of every chart shown in this work.

Have the authors examined the onset of mesodermal markers (CD13, ROR2, PDGFRa etc) in differentiating EOMES knockout cell lines? The microarray data (Table S1) shows that transcripts for PDGFRa are up-regulated in the TET-On EOMES samples whereas ROR2 levels do not vary. These data may be useful in quantifying mesodermal cell production in the various cell lines used.

In general, even primitive streak markers and early mesoderm genes are severely compromised in EOMES KO cells, as highlighted in the heat map representation of Fig. 2,e. Nonetheless, as correctly pointed out by R2, some cardiac-specific genes seem to be rather unaffected - ISL1 being one striking example (Fig. 2,e, Table S1,). Clearly, there is much room

here for follow-up investigation and it is for this reason that we decided to make the entire time-course expression table available as easily accessible supplementary information.

How many PiggyBac lines are used in the experiments? Is all the data from a single clone or are multiple lines used to confirm the findings?

In the revised manuscript, we have now clarified on several occasions which cell lines have been used. The study is based on three clonal cell lines - an EOMES^{KO/E.TET-ON} hESC line, a WT^{E.TET-ON} hESC line (both on HuES6 background), and a WT^{E.TET-ON} hiPSC line.

Figure 2 legend: “~1.5 weeks” and “~3 weeks” are somewhat vague. Please provide exact days of differentiation.

The symbols were to indicate that similar outcomes could independently be obtained "around" the indicated time-points. But we agree to the suggestion and have accordingly modified information in the figure and legend ("d 10" and "21 days").

The modified hESC lines were not karyotyped to ensure that no abnormalities arose during the genetic manipulation and no evidence is shown that they maintain pluripotency markers.

As suggested, we are now providing karyotype analyses for all three cell lines (in Fig. S2,a and S3,a,e; n = 10 each). In addition, Fig. S3,b,f show robust SOX2 expression in the undifferentiated state of 2 of these lines. Otherwise, we think that the numerous cardiac and non-cardiac differentiation experiments throughout this study inherently provide compelling evidence of the pluripotency of our cell lines.

Figure 3 please provide a schematic of the WNT3 and WNT3a alleles showing the region deleted and the sequence data demonstrating that the allele. Also, a second PCR should be performed using a primer internal to the CRISPR mediated deletion to confirm the knockout. The smaller PCR product may preferentially amplify meaning that a heterozygous genotype may be misclassified as a homozygote. An internal/external PCR combination should only produce a product if a wild type allele is still present. This could be presented in supplementary data.

Another useful suggestion. New Fig. S4,b contains a corresponding schematic of the knockout approach as well as the additional diagnostic PCR data. The corresponding primer and CRISPR sequences are listed in Table S3,a and can readily be employed to locate the corresponding sites in the genome using e.g. BLAT searches. In addition, we are now providing the genomic sequences in the WNT3 and WNT3A loci that have been deleted from the genome (in Table S3, b).

Figure 3d please indicate on the figure that the WNT inhib. was included with the BMP inhibitor. This experiment should also be repeated using the BMP inhibitor alone.

We have modified Fig. 4,b accordingly. The suggested experiment has been carried out (see above) but we do not wish to include it in the manuscript. Instead, in the legend to Fig. 4,b, we refer to a related experiment in Fig. 3,b.

The microarray data presented in Table S1 could be presented more effectively. A summary of the data for each transcript (rather than each spot on the microarray), including a column showing fold differences between the samples, should be included as a separate table within the excel spreadsheet. This would make it easier for the reader to identify which gene networks are dysregulated.

As suggested, we have now included an additional spreadsheet with expression ratios against the standard WT differentiation time-course (Table S1,b). Furthermore, it is true that many genes on the array are represented by more than one probe. In many cases, though, these are isoform-specific detection probes, which prohibits averaging the corresponding signal intensities. Instead, we have now included all detection probe sequences as an additional column in the supplementary table and this will enable the readers to selectively look up where they bind to.

The model in Fig. 3j suggests the EOMES-WNT form a cardiac mesoderm induction module, however, further data are required to confirm that this indeed cardiac specific mesoderm. Further differentiation of these cells along blood,

endothelial and other mesodermal lineages is required to formally establish that EOMES-WNT specific pre-cardiac mesoderm.

As strongly confirmed by the additional suggested experiments of Fig. 1_{r,d}, S1_r, S3_{r,c,d}, as well as by the new WT^{E.TET-ON} hiPSC-based data (Fig. 3_c and S3_{r,f}), efficient cardiac induction is confined to a cell clone-dependent DOX - i.e. EOMES expression level - range. Outside this range, EOMES drives differentiation into alternative fates. These results and observations are now incorporated into an extended model presented in Fig. 4_h. In essence, we do not claim that EOMES overexpression as such is cardiac-specific, which resembles the effects of BMP/WNT-activating molecules that are likewise not cardiac-specific.

Reviewer #3 (Remarks to the Author):

The role of Eomes in cell fate specification and in regulating WNT3 expression is novel and important. I find this elegant study very interesting and overall convincing and highly recommend publication after addressing the following points:

1. The treatment of WNT3 KO cells with CHIR compound in Fig.3i is a nice specificity control, but a more quantitative readout of the rescue of cardiac differentiation would strengthen the conclusion. In addition, Methods should state how many independent CRISPR clones were analyzed per genotype.

We were able to isolate one CRISPR clone per genotype, while given the dispensability of WNT3A in this context, we would argue that the WNT3 KO and the DKO could serve as specificity controls for one another. The CHIR supplementation experiments were performed separately from the primary experiments analyzed in Fig. 4,f, and by contrast, they were only analyzed in a semiquantitative manner, namely by estimating the percentage of beating cell surface area through a stereo microscope. Due to the semiquantitative nature of these data, we did not wish to incorporate them in the manuscript but I would like to emphasise that all cardiac immunostains shown in this work were overall representative ones (as also pointed out in the legend of Fig. 4,g). Given the high overall workload imposed in this revision, we hope R3 does not insist on repeating these demanding CHIR supplementation experiments, which requires a perfect control of culture conditions and synchronisation of four different cell lines in parallel:

The CHIR dose required for achieving a full rescue here was rather high (8 μM - indicated in legend to Fig. 4,g). Interestingly, though, this roughly corresponds to the concentration range used in standard "CHIR-only" cardiac differentiation protocols (e.g. Lian et al., PMID 22645348) and suggests that there is rather strong autocrine WNT3 signaling in WT^{E.TET-ON} cells.

2. Please indicate the statistical significance of the GO term enrichments shown in Fig. 1d.

The statistical significance of the data in Fig. 1,d is implied in the bar values themselves which denote "-log₁₀ of the p value" (a bar value of 10 would denote a p value of 10⁻¹⁰, for instance). For clarification, we have accordingly expanded the corresponding figure legend in the revised manuscript.

3. The legend of Suppl. Fig. 2a states that n was 2 to 5. To estimate the significance of the cumulative effect on cardiac fate, the authors should state which differences did or did not reach statistical significance.

We agree and have accordingly performed 1-sided t-tests against the best condition (with plausible outcomes considering the inter-experimental variation and number of replicates). This is additionally provided for several pairwise comparisons in Fig. S2,c.

4. In Suppl. Fig. 2c, 'time ---- 50 ms' probably denotes the scale of the x-axis, but this should be clarified (ideally by drawing an x-axis with ticks).

In the revised version of the manuscript, we have replaced those data by a new experiment but have followed R3's recommendation when drawing the axis.

5. A description of Fig. 3e seems to be missing in the text.

This is now included (new Fig. 4,c / page 5). Thanks a lot for pointing this out.

Reviewers' Comments:

Reviewer #1:

Remarks to the Author:

The revised manuscript by Pfeiffer and colleagues addresses my initial concerns. Using an additional hiPSC line provides some assurance that the findings are more generalizable. The dox concentration-dependent induction of EOMES with the demonstration of alternative lineages also strengthen the findings. Overall, many of these findings are novel and add meaningfully to the literature by defining the critical role of EOMES in cardiogenesis.

Reviewer #2:

Remarks to the Author:

To address the reviewers concerns the authors present a large collection of new data and have altered the text where appropriate. The majority of the reviewers concerns have been adequately addressed.

However, the key claim that EOMES is a context-dependant cardiac master regulator remains contentious. This group had previously shown that EOMES is essential for the formation of cardiac competent mesoderm, in part via the repression of SOX2 (Rao Cell Stem Cell 2015). Further, EOMES has been shown in animal models to be essential for a range of mesendodermal lineages, and in the context of heart development a direct regulator of MESP1 (van den Aamele, EMBO Rep. 2012). In the revised manuscript the authors present evidence that in EOMES is required for endoderm differentiation and EOMES knockout cells readily differentiate down neuroectodermal lineages. In this respect, the title implying that EOMES is a context dependant cardiac master regulator neglects a fundamental requirement for EOMES in endoderm development. The over-expression data using the TET-ON system demonstrate that EOMES is capable of patterning mesoderm in a dose dependent manner. The title should reflect this finding rather than the cardiac using the term cardiac master regulator. Cardiac differentiation in this system still relies up a subsequent suppression of WNT signalling, if this suppression is replace by other signalling (eg VEGF) a different cell population would be generated. Historically, a master regulator is a transcription factor that single handedly converts cell fate, eg. ANTP or MYOD. This is not the case for the data presented in this manuscript. This reviewer feels that this data support the hypothesis that EOMES is critical for the specification of cardiac mesoderm but the term "cardiac master regulator" could be misleading.

Other issues

As raised previously, the cumulative bar graphs are extremely difficult to interpret and relative expression is used without clearly stating what the values are relative too. For example, in figure 1 c expression of NKX2-5 is given as 3. Firstly, the Y-axis says "rel. expr. on day 8 [max=1]". So if max=1, how can NKX2-5 expression equal 3? Secondly, does this mean that NKX2-5 expression is 3 fold higher than CTNT and 1.5 higher than MYH6? This data is completely inconsistent with the known difference in mRNA levels of transcription factors and sarcomeric proteins. By contrast, in Figure 3c MYH6 is now 3 fold higher than NKX2-5 only 1 day earlier in differentiation. Is this due to line specific differences or is this difference due to the widespread over expression of TET-EOMES or is it due to the timing of Q-PCR? Please present this data in a more appropriate manner for the reader understand.

The authors show "unbiased expression analysis of meso-permissive differentiation" in Figure 1d, however, gene expression data underlying this is not shown. This data should be included. Only select GO annotations are shown and the data for this GO annotation is not presented. Until this data is shown it remains possible that these genes are present in multiple GO terms and that those

selected are not mesodermally restricted. The data in Figure 2e clearly show that mesoderm development is dramatically impaired in EOMES knockouts with dramatic reductions in MESP1, MIXL1, T and LHX1. Supporting the hypothesis that this EOMES is required for mesoderm formation rather than cardiac mesoderm specifically.

Reviewer #3:

Remarks to the Author:

The authors have satisfactorily addressed my comments, except for the first: I could still not find information in the Methods that only one clone per genotype was analyzed. I think this info is important for the readers. I also recommend to include (line 244?) how many correctly targeted clones were obtained: This is useful for anyone seeking to reproduce this targeting experiment.

We wish to thank the reviewers once more for reviewing the changes made to the manuscript. In this second round, we have now addressed the remaining open points as detailed below. Changes to the manuscript text are highlighted in blue font colour.

REVIEWERS' COMMENTS:

Reviewer #1 (Remarks to the Author):

The revised manuscript by Pfeiffer and colleagues addresses my initial concerns. Using an additional hiPSC line provides some assurance that the findings are more generalizable. The dox concentration-dependent induction of EOMES with the demonstration of alternative lineages also strengthen the findings. Overall, many of these findings are novel and add meaningfully to the literature by defining the critical role of EOMES in cardiogenesis.

Reviewer #2 (Remarks to the Author):

To address the reviewers concerns the authors present a large collection of new data and have altered the text where appropriate. The majority of the reviewers concerns have been adequately addressed.

However, the key claim that EOMES is a context-dependant cardiac master regulator remains contentious. This group had previously shown that EOMES is essential for the formation of cardiac competent mesoderm, in part via the repression of SOX2 (Rao Cell Stem Cell 2015). Further, EOMES has been shown in animal models to be essential for a range of mesendodermal lineages, and in the context of heart development a direct regulator of MESP1 (van den Aamele, EMBO Rep. 2012). In the revised manuscript the authors present evidence that in EOMES is required for endoderm differentiation and EOMES knockout cells readily differentiate down neuroectodermal lineages. In this respect, the title implying that EOMES is a context dependant cardiac master regulator neglects a fundamental requirement for EOMES in endoderm development. The over-expression data using the TET-ON system demonstrate that EOMES is capable of patterning mesoderm in a dose dependent manner. The title should reflect this finding rather than the cardiac using the term cardiac master regulator. Cardiac differentiation in this system still relies up a subsequent suppression of WNT signalling, if this suppression is replace by other signalling (eg VEGF) a different cell population would be generated. Historically, a master regulator is a transcription factor that single handedly converts cell fate, eg. ANTP or MYOD. This is not the case for the data presented in this manuscript. This reviewer feels that this data support the hypothesis that EOMES is critical for the specification of cardiac mesoderm but the term "cardiac master regulator" could be misleading.

In a strict sense, R2 is right. The function of EOMES to program hPSCs into a cardiogenic fate may not apply to other starting cell types such as fibroblasts. It appears to us, though, that nowadays the term "master regulator" is also being used in a more general way, to include powerful transcription factors capable of inducing entire developmental programmes as in the present case. As an example, Mesp1 as well has been proposed to be a master regulator in the cardiac context, albeit perhaps somewhat prematurely (Bondué et al., 2008, Cell Stem Cell 3, 69-84). So, it seems that the definition of the term has loosened over time.

Nonetheless, to acknowledge the criticism (and to hopefully prevent further delays), we have now replaced the term "master regulator" by other expressions throughout - except for one occurrence in the discussion where we, however, toned down the statement ("may be considered"). In addition, we have completely revised the title. This latter modification was also to avoid the impression that EOMES activity would exclusively produce cardiac outcomes. In fact, in abstract, introduction, as well as in results (Fig. 1d), we fully recognise that EOMES is essential for endoderm differentiation as well. Otherwise, we think the data added in the first revision strongly support our model according to which diverse differentiation outcomes form in an EOMES expression strength-dependent manner.

Other issues

As raised previously, the cumulative bar graphs are extremely difficult to interpret and relative expression is used without clearly stating what the values are relative too.

For example, in figure 1c expression of NKX2-5 is given as 3. Firstly, the Y-axis says “rel. expr. on day 8 [max=1]”. So if max=1, how can NKX2-5 expression equal 3? Secondly, does this mean that NKX2-5 expression is 3 fold higher than CTNT and 1.5 higher than MYH6? This data is completely inconsistent with the known difference in mRNA levels of transcription factors and sarcomeric proteins. By contrast, in Figure 3c MYH6 is now 3 fold higher than NKX2-5 only 1 day earlier in differentiation. Is this due to line specific differences or is this difference due to the widespread over expression of TET-EOMES or is it due to the timing of Q-PCR? Please present this data in a more appropriate manner for the reader understand.

Now we understand where the misunderstanding lies - thanks very much for explaining. qPCR data may be normalized to any sample in a given experiment. In many cases, we choose to normalize the expression levels - gene-by-gene - to the sample which shows the highest expression level for a given gene in the experiment ("max"). In Fig. 1c, therefore, all three genes of the WT sample are set to a value of 1. Because we assay 3 genes, piling up the data (1+1+1) yields a value of 3 on the y axis for the WT cells. The KO values are simply relative to that. Like with expressing pPCR data as fold changes against a certain sample, the values of different genes do not reflect differences at the absolute expression scale.

As said before, we wish to stick to this way of presenting the data because it significantly reduces complexity while preserving a high information content and robustness. However, we also acknowledge that additional explanation may be useful. Hence, we have modified the axis labelling and chart of Fig. 1c to better indicate how the data was normalised. In addition, we have now added an explanatory statement to the figure legend. We strongly hope R2 considers these measures sufficient.

The authors show “unbiased expression analysis of meso-permissive differentiation” in Figure 1d, however, gene expression data underlying this is not shown. This data should be included. Only select GO annotations are shown and the data for this GO annotation is not presented. Until this data is shown it remains possible that these genes are present in multiple GO terms and that those selected are not mesodermally restricted. The data in Figure 2e clearly show that mesoderm development is dramatically impaired in EOMES knockouts with dramatic reductions in MESP1, MIXL1, T and LHX1. Supporting the hypothesis that this EOMES is required for mesoderm formation rather than cardiac mesoderm specifically.

The underlying microarray experiment was a rather rough one - based on analyzing end point samples differentiated based on arbitrarily under or overdosing otherwise cardiac-promoting signaling factors. This was mainly to identify mesodermal marker genes in an unbiased manner - to then confirm these at the protein level, which is shown in previously revised Supplementary Fig. 1. We think these data provide compelling evidence that EOMES disruption does not disable mesodermal differentiation capability in general. Admittedly, in light of the in vivo Eomes KO phenotype this may seem a bit surprising. We think the explanation may reside in the fact that signaling activity may be added exogenously in a cell culture setting, i.e. the cultured cells do not fully depend on producing these cues in an autocrine fashion - in contrast to Eomes KO embryos where our model would predict a severe lack of autocrine Wnt(3) activity.

We concur with the reviewer that showing the actual expression data in addition to the immunostainings of Suppl. Fig. 1 may further increase the credibility of our data. Hence, the filtered expression data set underlying the GO analysis of Fig. 1e is now shown in new Supplementary data file 1.

Reviewer #3 (Remarks to the Author):

The authors have satisfactorily addressed my comments, except for the first: I could still not find information in the Methods that only one clone per genotype was analyzed. I think this info is important for the readers. I also recommend to include (line 244?) how many correctly targeted clones were obtained: This is useful for anyone seeking to reproduce this targeting experiment.

We regret not having provided numbers associated with the gene disruption experiments before. This information is now given in the "genetic manipulation" section in methods (24-32 clones screened per knockout, 1-4 positives obtained). We also state in there that one clone per genotype was analyzed in the functional experiments. (As indicated before, the DKO cells behave like the WNT3 KO ones, such that they validate one another.)